# A Novel Binary Hybrid PSO-EO Algorithm for Cryptanalysis of Internal State of RC4 Cipher

**DOI:** 10.3390/s22103844

**Published:** 2022-05-19

**Authors:** Rizk M. Rizk-Allah, Hatem Abdulkader, Samah S. Abd Elatif, Wail S. Elkilani, Eslam Al Maghayreh, Habib Dhahri, Awais Mahmood

**Affiliations:** 1Department of Basic Engineering Science, Faculty of Engineering, Menoufia University, Shebin El-Kom 32511, Menoufia, Egypt; rizk_masoud@sh-eng.menofia.edu.eg; 2Department of Information Systems, Faculty of Computers and Information, Menoufia University, Shebin El-Kom 32511, Menoufia, Egypt; hatem.abdelkader@ci.menofia.edu.eg; 3Department of Basic Engineering Science, Higher Institute of Engineering and Technology, Tanta 31739, Gharbia, Egypt; 4College of Applied Computer Science, King Saud University, Riyadh 11451, Saudi Arabia; ealmaghayreh@ksu.edu.sa (E.A.M.); hdhahri@ksu.edu.sa (H.D.); mawais@ksu.edu.sa (A.M.); 5Department of Computer Science, Yarmouk University, Irbid 21163, Jordan; 6Faculty of Sciences and Technology, University of Kairouan, Sidi Bouzid 4352, Tunisia

**Keywords:** cryptanalysis, known plaintext attack, stream cipher, hybrid binary optimization, fitness function, Particle Swarm Optimization

## Abstract

Cryptography protects privacy and confidentiality. So, it is necessary to guarantee that the ciphers used are secure and cryptanalysis-resistant. In this paper, a new state recovery attack against the RC4 stream cipher is revealed. A plaintext attack is used in which the attacker has both the plaintext and the ciphertext, so they can calculate the keystream and reveal the cipher’s internal state. To increase the quality of answers to practical and recent real-world global optimization difficulties, researchers are increasingly combining two or more variations. PSO and EO are combined in a hybrid PSOEO in an uncertain environment. We may also convert this method to its binary form to cryptanalyze the internal state of the RC4 cipher. When solving the cryptanalysis issue with HBPSOEO, we discover that it is more accurate and quicker than utilizing both PSO and EO independently. Experiments reveal that our proposed fitness function, in combination with HBPSOEO, requires checking 10^4^ possible internal states; however, brute force attacks require checking 2^128^ states.

## 1. Introduction

Cryptology is the study of methods that ensure information secrecy. Cryptology has two branches: cryptography and cryptanalysis. Cryptography studies the design of cryptosystems, whereas cryptanalysis studies the cracking of crypts or cryptosystems for retrieving crucial information. Cryptanalysis is the science of deciphering ciphertext to obtain its plaintext without knowing the secret key employed. Any cryptosystem’s cryptanalysis can be stated as an optimization problem.

The work of cryptanalysis is difficult. A cryptanalyst can use a number of different methods to break a cipher, depending on the amount of information available to the attacker. The Known Plaintext Attack (KPA) is a type of attack where the attacker possesses plaintext and matching ciphertext samples [1].

The Ciphertext Only Attack (COA) is another form of attack where the cryptanalyst knows the ciphertext only. The KPA is easier to implement than the COA because the attacker has access to more information (both plaintext and ciphertext pairs), allowing the secret key to be extracted more readily.

The two main components of modern cryptography are symmetric-key cryptography and asymmetric-key cryptography. The first group is then subdivided into block and stream ciphers as shown in (Figure 1). The operational premise of a stream cipher is that for every bit of plain text, there is a corresponding bit of keystream; these two are combined with an XOR operation to obtain one bit of ciphertext. The fundamental issue is that each piece of plaintext is coded independently from the others. Stream ciphers come in a variety of shapes and sizes, and they are useful in real-world applications such as: CSS used in DVD encryption, A5/1 and A5/2 used in GSM encryption, EO used in Bluetooth and RC4 used in SSL and WEP applications [2].

Heuristics is a technique used in computer science, artificial intelligence, and mathematical optimization to solve a problem faster when traditional techniques are too slow, or to discover a rough answer when traditional approaches fail to provide a perfect solution. A metaheuristic is an iterative strategy that drives a subordinate heuristic by intelligently mixing several notions for exploring and utilizing the search space. These algorithms are inspired by natural events. Meta-heuristic optimization algorithms are gaining popularity in engineering applications because they: (i) rely on relatively simple concepts and are easy to implement; (ii) do not require gradient information; (iii) can bypass local optima; and (iv) can be used to a wide range of challenges in a variety of fields. Meta-heuristic algorithms inspired by nature mimic biological or physical occurrences to solve optimization issues. They have been classified into four categories (see Figure 2): evolution-based, physics-based, swarm-based and human-based algorithms. The laws of natural evolution motivate evolution-based approaches. Genetic Algorithms (GA) [3], which replicate Darwinian evolution, are the most prominent evolution-inspired approach. Evolution Strategy (ES) [4], Genetic Programming (GP) [5], and Biogeography-Based Optimizer (BBO) [6] are some other popular algorithms. Physics-based methods, such as Simulated Annealing (SA) [7], Gravitational Search Algorithm (GSA) [8] and Equilibrium optimizer (EO) [9], emulate the physical universe’s principles. Swarm-based strategies, which mirror the social behavior of groups of animals, make up the third category of nature-inspired methods. Particle Swarm Optimization [10], Ant Colony Optimization (ACO) [11], Grey Wolf Optimization (GWO) [12] are the most popular algorithms in this category. Other meta-heuristic methodologies inspired by human behavior in the literature are also available, such as Teaching Learning Based Optimization (TLBO) [13], Harmony Search (HS) [14] and Tabu (Taboo) Search (TS) [15].

The goal of all of these algorithms is to discover the best solution for quality and convergence as efficiently as possible. A nature-inspired algorithm should incorporate exploration and exploitation elements to guarantee that the global optimum is discovered. Exploration is defined as the capacity to search the universe at large. This capacity is linked to escaping local optima and avoiding stagnation in local optima. Exploitation, on the other hand, refers to the capacity to look for potential ideas locally and increase their quality. Good performance is the outcome of a good trade-off between these two attributes. It is tough to strike a balance between these two qualities because when one improves, the other deteriorates [16]. A hybrid technique is one method for establishing a good balance, in which two or more algorithms are combined to make each algorithm’s performance better; the resulting hybrid technique is referred to as a memetic method [17].

It is not a novel idea to use metaheuristic optimization to tackle cryptanalysis challenges. Since the early 1990s, there has been a lot of research carried out on this subject, although the scope has been limited to a small number of cryptography and optimization methods. We will discuss a novel technique in this study that combines PSO and EO algorithms to increase their performance, and we will apply this approach to the cryptanalysis problem. The remainder of this work is broken down into the sections below: Section 2 discusses the scope of work relating to the use of evolutionary approaches in the field of cryptanalysis. The mathematical models of PSO, EO and HPSO-EO are explained in Section 3. Section 4 introduces the stream ciphers and RC4 cipher. Section 5 and Section 6 introduce the fitness function and proposed algorithm “BHPSO-EO’’ for attacking the stream cipher (RC4), while Section 7 offers the results and discussion. Finally, in Section 8, the work’s conclusion is presented.

## 2. Related Work

Several prior studies have investigated RC4 cryptanalysis. In 1995, Roos [18] identified a link between the used key’s first three bytes and the RC4 keystream’s first byte. The RC4 creates a keystream that is biased in different methods to different sequences. In a correlation attack, Mantin and Shamir [19] demonstrated that the second output byte was skewed toward zero with a probability of 1/128, although it should be 1/256. When the original state’s third byte is 0 and the second byte is not equal to 2, the second output byte is always 0. Klein [20] published a cryptanalysis of the RC4 Cipher in 2008, which revealed more correlations between the output key sequence and the applied keys.

Metahuristic algorithms have already been used to attack stream ciphers; Bogdan and Călin [21] submitted a paper in 2011 using the Tabu Search Methodology to recreate the internal state of the RC4 stream cipher. In 2015, Iwona and Mariusz [22] applied the Genetic Algorithm to the cryptanalysis of the RC4 cipher. In this research, GA was applied to find the keystream for RC4 with different key lengths key lengths of 64, 80, and 128 bits. We found that in the three different experiments, the fitness did not reach one, which indicates that the key obtained by the algorithm is an approximate key and is not real; however, in our paper, we obtain a real key as we obtained a value of fitness of one, which means that all the bits of the keystream are true, indicating that the proposed algorithm is more accurate. In 2019, Maiya, Saibal and Muttoo applied Particle Swarm Optimization to solve the RC4 stream cipher. The authors used COA in cryptanalysis to find the initial value of the key and not the keystream, whereas this paper focuses on finding the keystream in RC4 using KPA; we also introduce a novel algorithm based on the hydride PSO and EO that enhances their performance.

Various researchers have used hybrid algorithms to handle a range of problems in the optimization field during the last few decades. In solving numerous complex problems, these hybrid algorithms have outperformed their equivalents [23]. In [24], GA was combined with PSO to achieve global optimization. The authors of this study used GA and PSO hybridization to generate people not just from crossover and mutation operators, but also from PSO’s global and local search operators. For global numerical optimization, GA has been combined with the Taguchi method [25]. In [26], the authors of this study used hybrid Ant Colony Optimization with Firefly Algorithm for unconstrained optimization problems. In [27], GWO was hybridized with CSA based on the dynamic fuzzy learning strategy to solve large scale optimization problems. Hybrid strategies did substantially better in these tests than other global or local search approaches.

## 3. Methods

### 3.1. Particle Swarm Optimization

The Particle Swarm Optimization approach (PSO) was invented in 1995 by James Kennedy and Russell C. Eberhart [28]. This algorithm used to solve nonlinear optimization problems. Simulations of social psychological expression in birds and fish inspired this method. In the classic PSO model, a swarm of particles is given a population of random alternative solutions. They are constantly looking for new solutions in the D-dimension problem space. The fitness of a particle is directly proportional to its position. At time *t*, particle *j*th has a velocity ($v_{ij}{}^{t}$) and a location ($x^{t}$). The terms *P* best and *G* best are included in PSO’s equation. Over the course of each iteration, these mathematical equations are utilized to update position and velocity:(1)$$v_{ij}{}^{t+1}=wv_{ij}{}^{t}+c_{1}R_{1}\left(Pbest^{t}-x^{t}\right)+c_{2}R_{2}\left(Gbest^{t}-x^{t}\right)$$
(2)$$x^{t+1}=x^{t}+v^{t+1},\;\left(i=1,2,\ldots \;\ldots np\right)and\;\left(j=1,2,\ldots \;\ldots nv\right).\;$$
where
(3)$$w=\frac{w^{max}-w^{min}*iteration}{maxiteration}$$
where *w* is the inertia weight. $Pbest^{t}$: among all particles, this is the best location of all time $Gbest^{t}$: among all particles, this is the best global location. Figure 3 shows the flow chart for it.

$w^{max}=0.4\;,\;w^{min}=0.9\;,v_{ij}{}^{t}$,$\;v_{ij}{}^{t+1}$ is the velocity of the *j*th member of the *i*th particle at iteration number (*t*) and (*t* + 1). Usually, *R*_1_ and *R*_2_ are random numbers between (0, 1), $c_{1}=c_{2}=2$.

### 3.2. Equilibrium Optimizer (EO)

The Equilibrium Optimizer (EO) [9] is a novel continuous optimization approach based on physical principles. Because of its excellent exploration and exploitation capabilities, the EO technique has the benefit of being able to modify the answer at random. Particles with EO concentrations are similar to the particles and locations used to represent search agents in Particle Swarm Optimization (PSO). The search agents alter their concentration at random with regard to the best-so-far solutions, namely equilibrium candidates, to eventually achieve the equilibrium state (optimal outcome). The following is a more in-depth description of EO’s central concept.

#### 3.2.1. Inspiration

The EO algorithm takes its inspiration from the physical mass balance equation. The mass balancing equation provides a physical basis for the conservation of entering mass, departing mass, and generated mass in the control volume. A first-order ordinary differential equation represents a universal mass balancing equation, in which the change in mass over time equals the mass entering the system minus the mass departing the system plus internal generation. The mass balancing equation is represented by Equation (4).
(4)$$v\frac{dc}{dt}=Q\;C_{eq}-QC+G\;$$
where v denotes the control volume, C denotes the concentration of particles in the control volume, $v\frac{dc}{dt}$ denotes the control volume’s mass change rate, Q denotes the flow rate of volumetric fluid into and out of the control volume, $C_{eq}$ denotes the the particle density within the control volume at an equilibrium state without generation, and G denotes the mass generation rate within the control volume. A steady equilibrium state occurs when $v\frac{dc}{dt}$ = 0 and$\;\lambda =\frac{Q}{v}$ defined as turnover rate (the inverse of residence). Equation (4) can be rearranged to solve for $\frac{dc}{dt}$ as a function of $\frac{Q}{v}$.
(5)$$\frac{dc}{\lambda \;C_{eq}-\lambda \;C+\frac{G}{v}}=dt$$

Equation (6) shows the time-integration of Equation (5).
(6)$$\int_{C_{0}}^{C}\frac{dc}{\lambda \;C_{eq}-\lambda \;C+\frac{G}{v}}=\int_{t_{0}}^{t}dt$$

The following is the end result:(7)$$C=C_{eq}+\left(C_{0}-C_{eq}\right)F+\frac{G}{\lambda v}\left(1-F\right)$$
where *F* is the following formula:(8)$$F=e^{-\lambda \left(t-t_{0}\right)}$$

The beginning start time and concentration are $t_{0}$ and$\;C_{0}$, respectively, and are dependent on the integration interval.

The general structure of the EO technique is given by Equation (7). As Equation (7) shows, three terms have an impact on the EO algorithm’s search and update pattern. In the next part, we will go through the algorithm and update its formula in detail.

#### 3.2.2. Equilibrium Pool ($C_{eq})$

The final convergence state of the method, which is meant to be the global optimum, is called the equilibrium state [9]. The equilibrium pool is generated using the EO algorithm, which contains candidate particles for equilibrium. Experiments identified five candidate particles for the equilibrium pool, four of which were determined as the best particles during the optimization phase, and the fifth is the mathematical mean of the four above particles. Four best particles aid in bettering the exploration, the average, on the other hand, contributes to the exploitation. The equilibrium pool’s vector is as follows:(9)$${\vec{C}}_{eq,pool}=\left\{{\vec{C}}_{eq\left(1\right)},\;{\vec{C}}_{eq\left(2\right)},\;{\vec{C}}_{eq\left(3\right)},{\vec{\;C}}_{eq\left(4\right)},\;{\vec{C}}_{eq\left(ave\right)}\;\right\}$$

It is important to note that each iteration’s particle update is carried out using the random selection approach; as a consequence, all possible solutions for each individual update at around the same time.

#### 3.2.3. Exponential Term (F)

The exponential term (F) is crucial in balancing the EO algorithm’s exploration and exploitation. (F) is calculated according to Equation (8). The iterative function time (*t*) decreases as the number of iterations increases, and λ is a random vector between [0, 1].
(10)$$t={\left(1-\frac{iter}{Max-iter}\right)}^{\left(a_{2}\frac{iter}{Max-iter}\right)}$$

The current and maximum numbers of iterations are iter and Max−iter, respectively. In Equation (8), the value of $t_{0\;}$ is computed as follows:(11)$${\vec{t}}_{0}=\frac{1}{\vec{\lambda }}\ln(-a_{1}sign(\vec{r}-0.5)\;\left[e^{-\vec{\lambda }t}-1\right])+t$$

The constants $a_{1}$ and $a_{2}$ regulate the exploration and exploitation capacities, respectively. A higher value of $a_{1}$ indicates that the exploration capability is better and the exploitation capability is lower. The greater value of $a_{2}$ indicates the lesser exploration ability, the stronger the exploitation ability. The exploration and exploitation direction is indicated by sign(r−0.5). Equation (8) becomes as shown in Equation (12) after replacing the values of t and ${\vec{t}}_{0}$ as indicated in Equations (10) and (11).
(12)$$\vec{F}=a_{1}sign\left(\vec{r}-0.5\right)\;\left[e^{-\vec{\lambda }t}-1\right]$$

#### 3.2.4. Generation Rate (G)

The EO algorithm’s most important parameter is the generating rate, as it aids in providing a precise answer by enhancing exploitation. The generation rate is calculated as follows:(13)$$\vec{G}={\vec{G}}_{0}\;e^{-\vec{k}\left(t-t_{0}\right)}$$

${\vec{G}}_{0}\;$ is the starting value, while *k* is the decay constant (k=λ). As a result, the generation rate is finally expressed as follows:(14)$$\vec{G}={\vec{G}}_{0}\;\vec{F}$$
where:(15)$${\vec{G}}_{0}=\vec{GCP}\;\left({\vec{C}}_{eq}-\vec{\lambda }\;\vec{C}\right)$$
(16)$$\vec{GCP}=\{\begin{array}{c} 0.5\;r_{1}\;\;\;\;\;\;\;\;\;\;\;r_{2}\geq GP\\ 0\;\;\;\;\;\;\;\;\;\;\;\;\;\;\;\;\;\;r_{2}<GP\;\;\;\;\;\;\; \end{array}$$

The probability that the generating term contributes to the updating process is represented by *GCP*, which is known as the control parameter for the generation rate; the number of particles that use generation terms to update their state is determined by the probability of this contribution. Equation (16) yields *GCP*, where (*GP* = 0.5) is the probability of the generation, and its function is to strike a reasonable balance between exploration and exploitation. $r_{1}$ and $r_{2}$ are two random values in the range of [0, 1]. Finally, EO’s rule for updating is as follows:(17)$$C=C_{eq}+\left(C-C_{eq}\right)F+\frac{G}{\lambda v}\left(1-F\right)$$

The second term in the EO algorithm is in charge of performing a global search in the search space to identify an optimum location, while the third term aids in increasing the solution’s precision. It is also important to note that EO uses a particle’s memory conserving mechanism, which helps to boost the algorithm’s exploitation capabilities. The flow chart of the EO algorithm is depicted in Figure 4.

### 3.3. Proposed Algorithm (HPSO-EO)

A set of hybrid PSO-EO is a combination of separate PSO and EO. Hybrid algorithms are created so that the strengths of one algorithm compensate for the drawbacks of the other. As a result, the constant inertia weight, PSO, has the constraint of only covering a narrow search space for tackling higher-order or complex design problems. The EO algorithm, on the other hand, provides several benefits. Firstly, the use of exponential terms (F) allows EO to strike a balance between exploration and exploitation. Secondly, the exact solution is provided by employing the generation rate. Finally, using an average particle in the computation of an equilibrium pool aids in the discovery of unknown search areas during the initial iterations when the particles are far apart. Therefore, we introduce a new hybrid PSO-EO algorithm that combines the benefits of both.

The proposed algorithm corrects the PSO algorithm’s flaw and enhances the exploration phase, allowing it to test a large number of viable solutions. During this phase, the best particle’s position in the EO replaces the particle’s location attempting to discover the optimal solution to a complex nonlinear problem. The best EO algorithm solutions are regarded as one of the proposed algorithm’s solutions. So, the EO algorithm directs the PSO particles to the most effective solutions. In this algorithm the velocity equation and the position equation have been computed as shown: (18)$$v_{ij}{}^{t+1}=wv_{ij}{}^{t}+c_{1}R_{1}\left(Pbest^{t}-x^{t}\right)+c_{2}R_{2}\left(Gbest^{t}-x^{t}\right)+c_{3}R_{3}\left(EO\_best^{t}-x^{t}\right).$$
(19)$$x^{t+1}=x^{t}+v^{t+1}$$

## 4. Stream Ciphers (RC4)

A keystream, which is a bitstream, is produced by the majority of stream cipher algorithms. This keystream is then XOR-ed with plaintext, which is a message to be encrypted. The encrypted stream is referred to as “ciphertext”, Figure 5 describe the stream cipher diagram. The key is used as a seed (initial value) for the stream cipher algorithm; this value is not used directly. This method is intended to generate a keystream that is significantly longer than the key itself. The keystream length depends on the plaintext length, but the initial value (IV) is independent of the plaintext length, which might be the same or different. Another difference between the IV and the keystream is that the keystream must be kept secret to ensure the secrecy of the communication, whereas the IV can be made public. The ciphertext and keystream must be XOR-ed in order to decrypt the encrypted message [29]. RC4 is a variable key-size stream cipher that is commonly utilized in software implementation because of its efficiency. It is found in SSL/TLS (Secure Socket Layer/Transport Layer Security) standards, as well as WEP (Wired Equivalent Privacy) and email encryption software. Ron Rivest of RSA Security created RC4 in 1987. RC4 is made up of two parts: a key-scheduling algorithm (KSA) that takes the key K (typically 40–256 bits in size) as the input and a keystream generator (PRGA) that generates a pseudo-random output sequence.

### 4.1. The Key-Scheduling Algorithm

The RC4 stream cipher’s initial step is the KSA; the KSA generates the initial permutation S from {0, …, *N* − 1} given a (random) key of length *l* bytes. The length of the key is usually between 5 and 32 characters. The key length’s maximum value *N* is 256 bits. S is set to {*0*, …, *N* − 1} and then the key bytes are mixed in using Algorithm 1. The input of this algorithm is the initial value of the key usually between 40 bits and 256 bits and the output is the initial permutation *S*.
**Algorithm 1** Key-scheduling**for** 0≤i≤N−1  S[i]=i**end for**j=0**for** 0≤i≤N−1   j=(j+S[i]+key[i mod keylength])mod N  swap (S[i],S[j])**end for**

### 4.2. The Pseudo-Random Generation Algorithm

PRGA employs the permutation S created by KSA to generate a pseudo-random key sequence. *S*(*i*) and *S*(*j*) are added together, and then looking up their sum *(mod N)* in *S*, the output byte is calculated. In the encoding process, the calculated output byte is used as a key sequence (Algorithm 2).
**Algorithm 2** The Pseudo-Random Generation*i* = *j* = 0**Do** *Generation Out Key Sequence*  i=(i+1) modulo N  j=(j+S[i])modulo N  swap (S[i] and S[j])  Out−Byte=S[(S[i]+S[j]) modulo N]**while**
Required Key Seq. generated


## 5. Fitness Function

There is one more major issue that has to be resolved. It is the fitness function $f_{fit}$ that qualifies how good a solution is to compare solutions. In this article, we used a plaintext attack, so the plaintext and ciphertext are both accessible to the attacker. The attacker can calculate the keystream by using both plaintext and ciphertext. Both attacker and the cryptanalysis algorithm generate keystreams that are identical in length. After that, the keystreams are compared to one another. When the fitness value is equal to 1, it signifies that the examined bits streams and the stream created by the cryptanalysis method are 100% compatible. The fitness value ranges from 0 to (1) that indicates how many matching bits there are in the provided solution. The equations below demonstrate how to compute fitness.
(20)$$K_{m}=P_{n}⊕C_{n}$$
(21)$$f_{fit}=\frac{\#(K_{m}⊕{\overset{´}{K}}_{m})}{\left|K\right|}$$
where the number of zeros in $\left(K_{m}⊕{\overset{´}{K}}_{m}\right)$ are denoted by #, and ⊕ specifies the XOR operation. $K_{m}$ is the keystream analyzed by the attacker, ${\overset{´}{K}}_{m}$ is the keystream created by the cryptanalysis algorithm, |K| is the keystream length in bits, $\;P_{n}$ is the known plaintext and $C_{n}$ is the known ciphertext.

## 6. Proposed Algorithm for Attacking RC4 Cipher Using Binary HPSOEO

The proposed algorithm is explained in addition to how HPSOEO can be used to discover the keystream utilized in RC4 cipher encryption and decryption. The issue is effectively finding the keystream. All of the approaches discussed in Section 3 were created to solve problems that required continuous optimization; in the case of cryptanalysis, the problem involved binary data. As a consequence, we may compute the position vector using Equation (19) and convert the continuous values to binary values using this formula:(22)$$x_{d}\left(t+1\right)=\{\begin{array}{c} 1\;\;,\;\;if\;\;\;\;\;\;rand\leq sigmoid\left(X\right)\\ 0\;\;,\;\;\;\;\;\;\;\;\;\;\;\;otherwise\;\;\;\;\;\;\; \end{array}$$
where sigmoid (X) is defined in Equation (23) and $x_{d}\left(t+1\right)$ is the binary update location at iteration *t* in dimension *d*.
(23)$$sigmoid\left(X\right)=\frac{1}{1+e^{-\left(X/2\right)}}$$

In this approach, each particle represents an (*l*-bit) binary key, where *l* is the key’s length that was used in the encryption (because the RC4 cipher is a varying key-size crypto algorithm) to set up a particle with *n* particles ∈[0, 1]. To evaluate each particle’s fitness, we use Equation (22). The particle with the greatest fitness occupies the best position. The velocity and particle position are updated using Equations (18) and (22). The updated particle’s location is used to compute the fitness. The technique is then continued until the appropriate number of iterations has been reached. Algorithm 3 depicts an algorithm for locating the keystream using BHPSOEO. Figure 6 shows the flow chart of BHPSOEO algorithm.
**Algorithm 3** Keystream Using BHPSOEOInitialization (initialization a  , GP ,w, velocity,position and concentrations).Randomly initialize particle’s positions of n particles and concentrations ∈ [0, 1].Calculation the fitness values of search particles according to Equation (21).Determine pBest , GBest, Ceq1,Ceq2, Ceq3 and Ceq4  iter=1 
 **While** (iter<Max_iter)   **For**
i=1 to no of particles   Calculate the value of fitness for each particle according to Equation (21).    Update $p_{Best}\;,\;G_{Best},\;C_{eq1},C_{eq2},\;C_{eq3}\;\mathrm{and}\;C_{eq4}$
   Compute $C_{ave}=(C_{eq1}+C_{eq2}+C_{eq3}+C_{eq4})/4$ and $C_{eq-pool}$ according to Equation (9)   **End For**
 Computing memory saving memory if iter>1 and compute *t* according to Equation (10)   **For**
i=1 to no of particles   Choose one candidate at random from the equilibrium pool (vector).    Compute $\vec{F}\;$according to Equation (12)    Compute $\vec{GCP}$ according to Equation (16)    Update concentrations according to Equation (17)    Update the velocity according to Equation (18)    Update the position vector according to Equation (22)   **End For**  iter=iter+1
 **End while**

## 7. Experimental Set Up and Result

The results of the experiments are displayed in Figure 7 and Figure 8. Figure 8 shows the comparison between the BPSO, BEO and BHPSOEO algorithms in the internal state of RC4 streams for 64-bit text length, and the initial value of the key is 40 bits in different population sizes. Figure 8 shows the comparison between these algorithms in the internal state of cryptanalysis of the RC4 stream for 128-bit text length, and the initial value of the key is 64 bits in different population sizes. From our results, we notice that firstly, the performance of BHPSOEO is better than the performance of other algorithms in different population sizes and different text length sizes. Secondly, when we increase the population size, the average fitness of all algorithms increases. Thirdly, when we increase the text length, the search space increases.

The BPSO, BEO and BHPSOEO algorithms for attacking the internal state of RC4 are implemented using the MATLAB program. Table 1 shows the numerical results for cryptanalysis RC4 cipher using HBPSOEO, BEO and BPSO when the text length is 64 bits. Table 2 shows the parameters are used in this exprement. These algorithms were executed 50 times with different population sizes. In the case of the text length of 64 bits, we used *N* = (10, 20, 30, 40, 50) and when the text length was 128, we used *N* = (50, 60, 70, 80, 90, 100).

From the results, we notice that the BHPSOEO algorithm gives us the best performance. It is the only algorithm that gives us the real key when the fitness is equal to 1; this means that all the bits in the tried key are correct. However, the other algorithms give us the maximum value of the fitness but do not reach one; this means that some of the bits in the tried key are not correct.

When the text length is 128 bits, Table 3 presents the numerical results for cryptanalysis of the RC4 encryption using HBPSOEO, BEO, and BPSO. Table 4 shows the parameters are used in this experiment. We can see from the results that the BHPSOEO algorithm performs the best. When the fitness equals 1, it is the only algorithm that offers us the real key, which signifies that all bits in the tested key are correct. However, the other algorithms offer us the maximum fitness value but do not attain one, implying that some bits in the tested key are incorrect.

The Root Mean Square Difference (*RMSD*) approach method was used to compare HBPSOEO vs. BEO and HBPSOEO vs. BPSO in Figure 9 and Figure 10. *RMSD* is a formula for calculating the difference between two sets of data *x_1_* and *x*_2_ as shown in (23). The wider the gap between data sets, the higher the *RMSD* number.
(24)$$RMSD=\sqrt{\frac{\sum_{1}^{N}{\left(x_{1,i}-x_{2,i}\right)}^{2}}{N}}$$

The difference between BHPSOEO and BPSO is high and nearly constant for different sizes, as shown in the plots in Figure 9 and Figure 10, but the average difference between BEO and BHPSOEO is initially big and subsequently declines. This proves that BHPSOEO is more efficient for cryptanalysis than both BPSO and BEO.

## 8. Conclusions and Future Works

A novel hybrid evolutionary technique that combines Particle Swarm Optimization (PSO) and Equilibrium Optimizer (EO) is presented in this study to increase the performance of both PSO and EO. This algorithm can be converted to binary form and used to solve the problems of cryptanalysis. We all know that cryptanalysis is a difficult problem to solve and that it takes a long time. From the results of our research, we were able to solve the cryptanalysis problem in a short time compared to traditional methods.

We use this novel algorithm to recover the internal state of the RC4 cipher, and this can be improved; this algorithm gives a more accurate solution than other algorithms such as PSO and EO. Compared to traditional methods such as brute force attacks, to find the internal state of the key in the case of a text length of 64 bits, we need (2^64^ keys) to find the real one, and in the case of a text length of 128 bits, we need (2^128^ keys) to find the real one; however, when we use the proposed algorithm in the case of a text length of 64 bits, we can obtain the key after checking 1000 keys only, and in the case of a text length 128 bits, we can obtain the real key after checking 5000 keys only. This proves that the proposed algorithm is very efficient in solving the cryptanalysis problem.

Finally, because the fitness function used in our experiment is independent of the cipher under attack, this technique can be easily adapted to other recent block or stream ciphers. As a result, we may utilize this method to attack AES. Using other swarm intelligence technologies to attack AES is also part of our future study.

The pros and cons of our methodology can be summarized in Table 5:

## Figures and Tables

**Figure 1 sensors-22-03844-f001:**
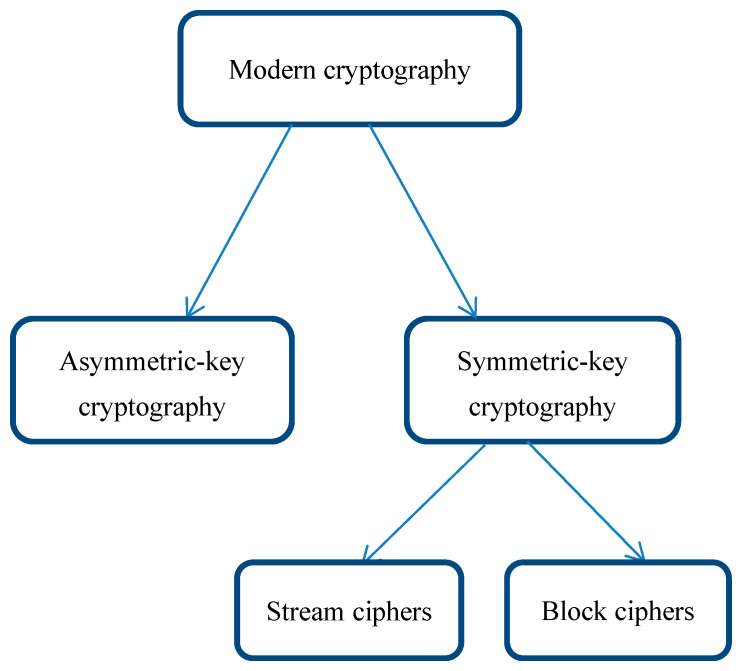
Main classification of a modern cipher.

**Figure 2 sensors-22-03844-f002:**
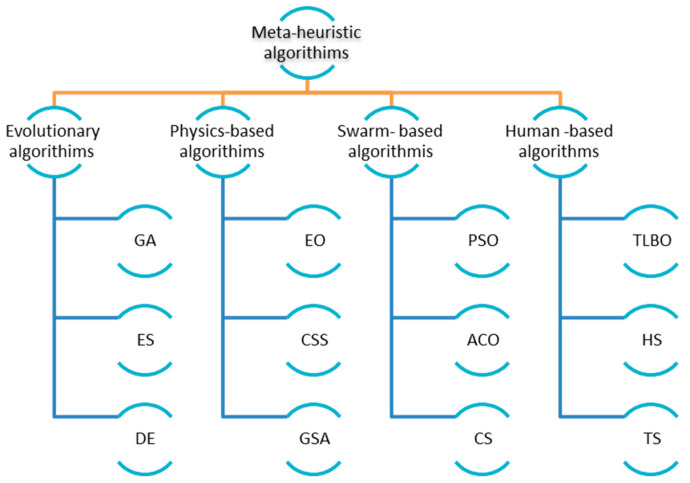
Classification of meta-heuristic algorithms.

**Figure 3 sensors-22-03844-f003:**
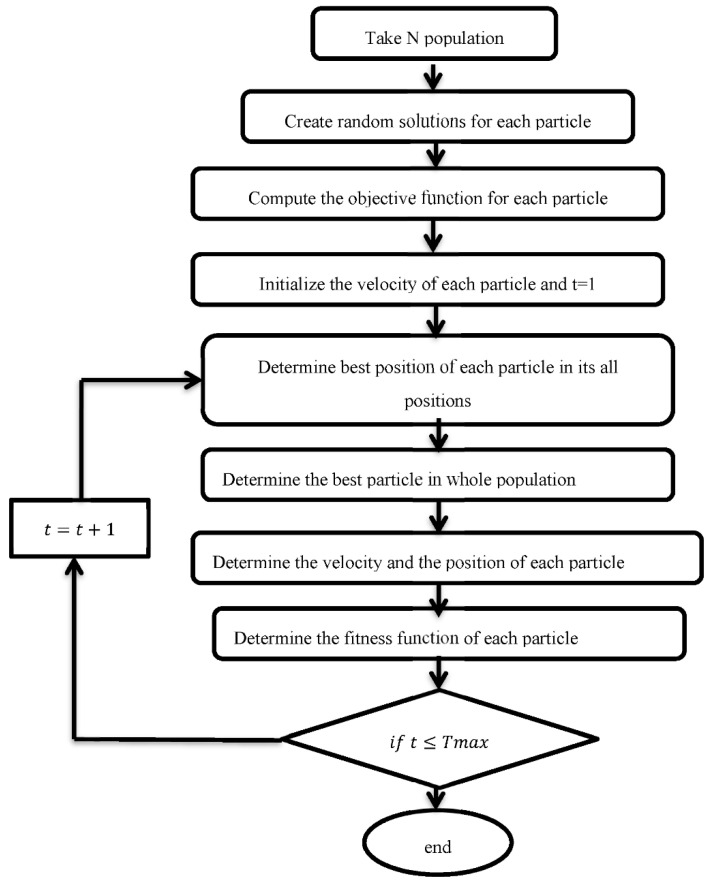
Flow chart of PSO algorithm.

**Figure 4 sensors-22-03844-f004:**
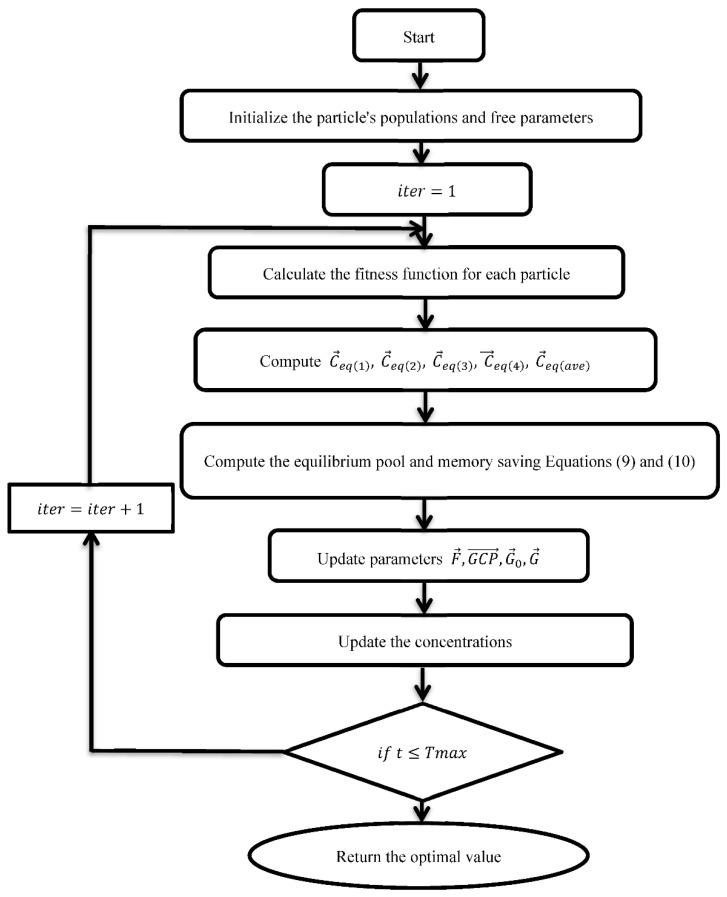
The flow chart of EO algorithm.

**Figure 5 sensors-22-03844-f005:**
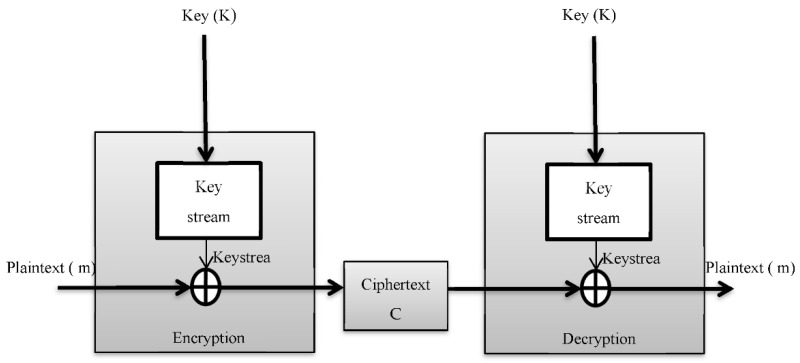
Stream cipher diagram.

**Figure 6 sensors-22-03844-f006:**
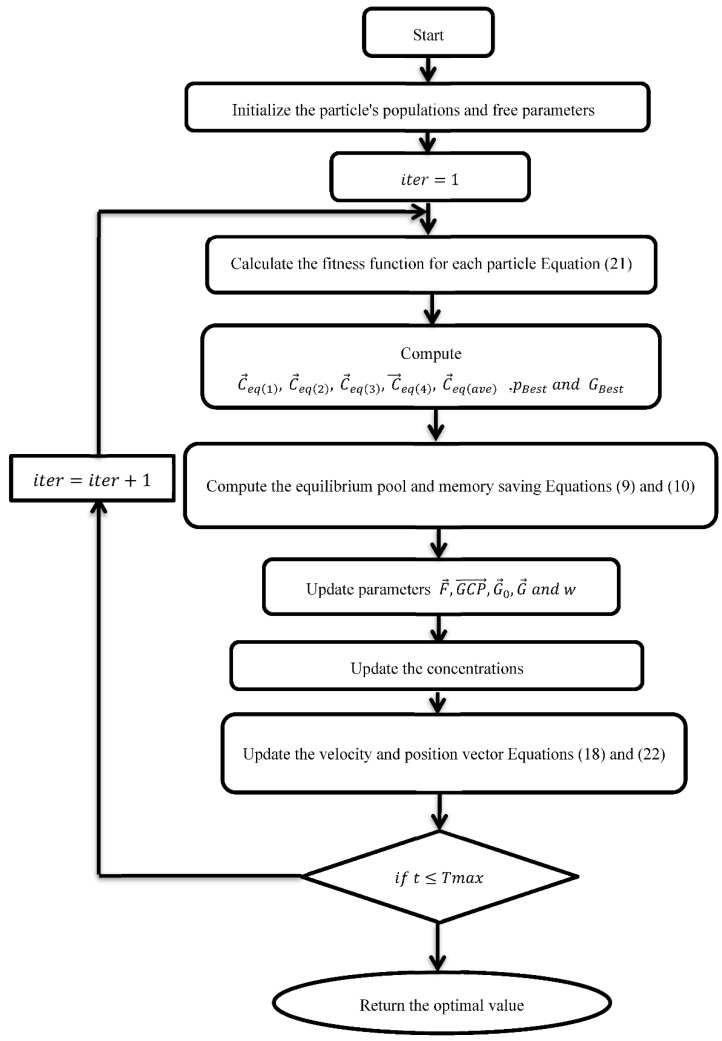
The flow chart of BHPSOEO algorithm.

**Figure 7 sensors-22-03844-f007:**
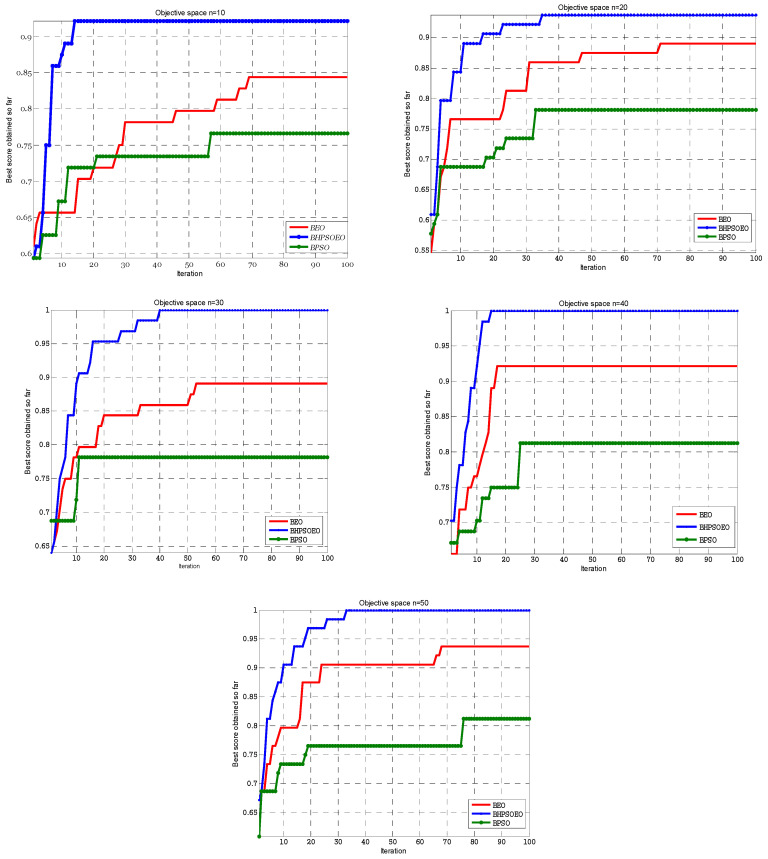
Convergence curve of BHPSOEO, BPSO and BEO for cryptanalysis of RC4 cipher with TextLen = 64 bits for different population sizes.

**Figure 8 sensors-22-03844-f008:**
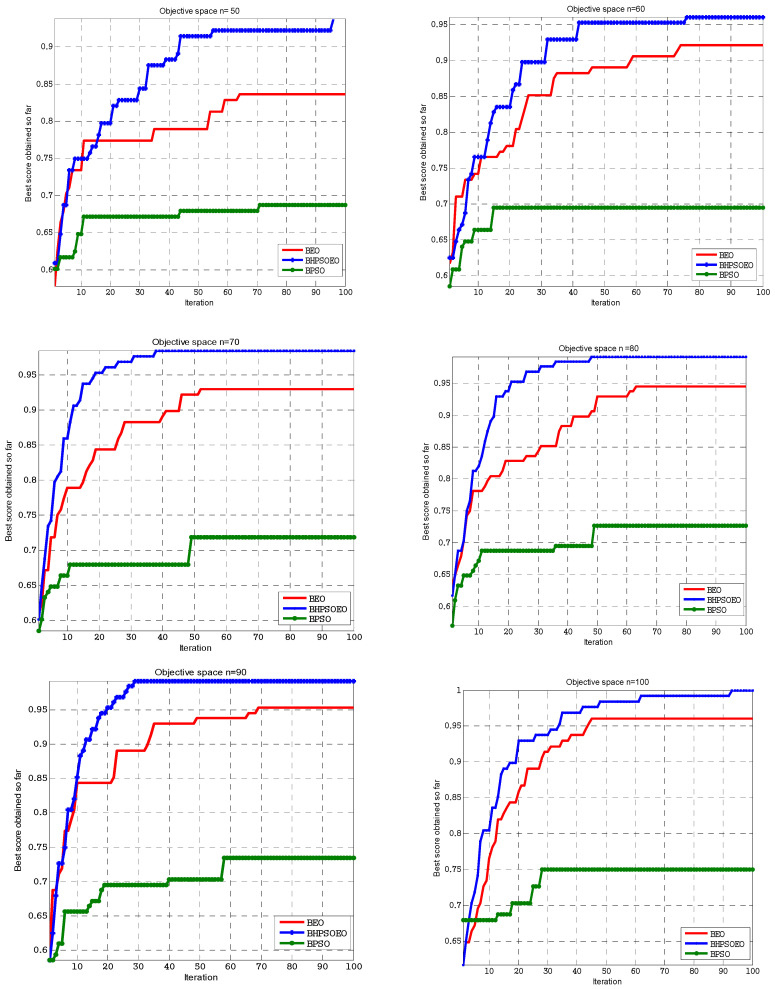
Convergence curve of BHPSOEO, BPSO and BEO for cryptanalysis of RC4 cipher with TextLen = 128 bits for different population sizes.

**Figure 9 sensors-22-03844-f009:**
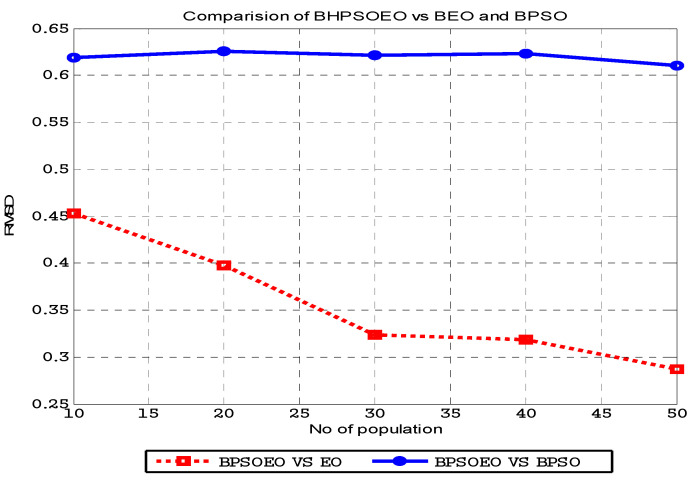
*RMSD* in average fitness for BPSOEO versus BEO and BPSO in cryptanalysis of the RC4 cipher when text length is 64 bits.

**Figure 10 sensors-22-03844-f010:**
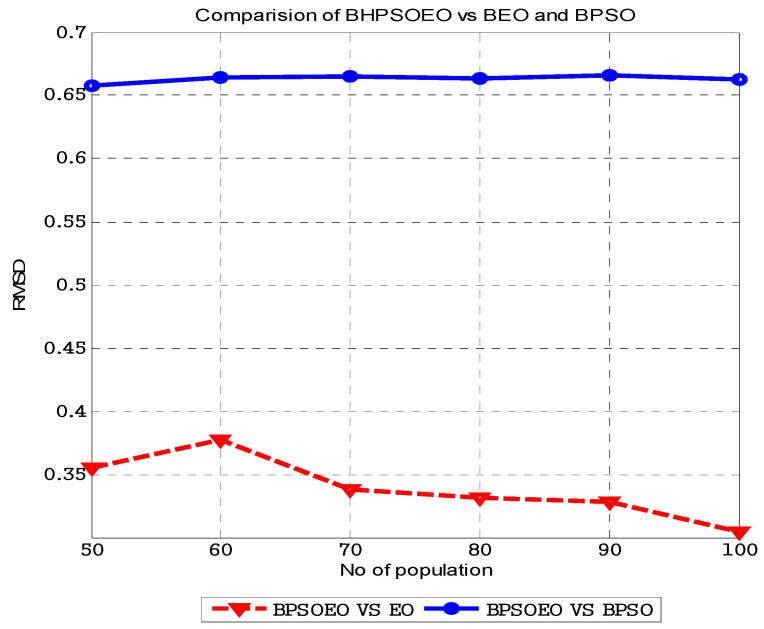
*RMSD* in average fitness for BPSOEO versus BEO and BPSO in the cryptanalysis of RC4 cipher when text length is 128 bits.

**Table 1 sensors-22-03844-t001:** BHPSEO, BEO and BPSO numerical results for cryptanalysis of RC4 cipher for a 64-bit text length.

		*N* = 10	*N* = 20	*N* = 30	*N* = 40	*N* = 50
BHPSOEO	Best fitness	1	1	1	1	1
	Worst fitness	0.82813	0.85938	0.90625	0.92188	0.92188
	The average	0.93719	0.97375	0.98469	0.99062	0.99094
	Std	0.03891	0.03911	0.02198	0.017576	0.014837
	Time	33.57 s	49.20341 s	99.58 s	100.37 s	123.895 s
BEO	Best fitness	0.90625	0.95313	0.96875	0.98438	0.98438
	Worst fitness	0.78125	0.84375	0.85938	0.82813	0.90625
	The average	0.85469	0.90281	0.92906	0.9425	0.95281
	Std	0.03765	0.027931	0.03064	0.028136	0.025153
	Time	15.58 s	30.80 s	46.33 s	65.17 s	60.66 s
BPSO	Best fitness	0.79688	0.7969	0.8125	0.82813	0.84375
	Worst fitness	0.70313	0.71875	0.73438	0.73438	0.75
	The average	0.74406	0.7625	0.76313	0.77563	0.78625
	Std	0.022739	0.01894	0.018504	0.021816	0.022419
	Time	12.30 s	35.83 s	50.61 s	67.25 s	86.99 s

**Table 2 sensors-22-03844-t002:** The BHPSOEO parameters to attack RC4 cipher with Textlen 64 bits.

Parameter	Definition	Value
c1	Cognitive Acceleration Coefficient	2.1
c2	Social Acceleration Coefficient	2.1
c3	Coefficient vector	2.1
W	Inertia Weight	0.9 + rand()/2
N	Population size	(10–50)
D	No of variables	64
K_L	The length of initial value of key	40
TexLen	The length of ciphertext and plaintext	64
Tmax	Maximum number of iterations	100
R	The number of times the algorithm is run	50

**Table 3 sensors-22-03844-t003:** BHPSEO, BEO and BPSO numerical results for cryptanalysis of the RC4 cipher of a 128-bit text length.

		*N* = 50	*N* = 60	*N* = 70	*N* = 80	*N* = 90	*N* = 100
BHPSOEO	Best fitness	1	1	1	1	1	1
	Worst fitness	0.88281	0.85156	0.89844	0.90625	0.90625	0.91406
	The average	0.95281	0.96344	0.96703	0.97188	0.97453	0.97188
	Std	0.031645	0.031585	0.030658	0.026222	0.022422	0.025204
	Time	127.03 s	105.28 s	176.86 s	201.98 s	225.69 0 s	246.83 s
BEO	Best fitness	0.95313	0.95313	0.96875	0.96875	0.97656	0.97656
	Worst fitness	0.85156	0.84375	0.875	0.86719	0.88281	0.88281
	The average	0.90891	0.90594	0.90281	0.92828	0.93078	0.9375
	Std	0.022906	0.023356	0.019799	0.021392	0.028136	0.020519
	Time	62.47 s	98.28 s	101.08 s	113.19 s	133.09 s	160.85 s
BPSO	Best fitness	0.75781	0.75	0.75	0.75781	0.75781	0.75781
	Worst fitness	0.69531	0.70313	0.70313	0.70313	0.71094	0.71094
	The average	0.72109	0.72281	0.72453	0.72906	0.72828	0.72953
	Std	0.014139	0.012045	0.011685	0.013979	0.011193	0.011148
	Time	85.93 s	87.30 s	90.41 s	101.33 s	110.85 s	135.49 s

**Table 4 sensors-22-03844-t004:** The BHPSOEO parameters to attack the RC4 cipher with Textlen, 128 bits.

Parameter	Definition	Value
c1	Cognitive Acceleration Coefficient	2.1
c2	Social Acceleration Coefficient	2.1
c3	Coefficient vector	2.1
W	Inertia Weight	0.9 + rand()/2
N	Population size	(50–100)
D	No of variables	128
K_L	The length of initial value of key	64
TexLen	The length of ciphertext and plaintext	128
Tmax	Maximum number of iterations	100
R	The number of times the algorithm is run	50

**Table 5 sensors-22-03844-t005:** The pros and cons of our methodology.

The proposed algorithm (BHPSOEO)	The pros	The cons
	The BHPSOEO algorithm gives us the best performance. It is the only algorithm that gives us the real key one when the fitness is equal to 1; this means that all bits in the tried key are correct. By comparing the proposed algorithm with the brute force attack in cryptanalysis of RC4, we find that if the key length is 64 bits, the search space is reduced from 2^64^ to 10^3^, and in the case where the key length is 128 bits, the search space is reduced from 2^128^ to (5 × 10^3^) More accurate than using GA [22] to attack RC4.	From the view of the running time, the BHPSOEO algorithm computation time is greater than the PSO and EO after the same iterations.

## Data Availability

Not applicable.

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
