# Peer review of "A Novel Binary Hybrid PSO-EO Algorithm for Cryptanalysis of Internal State of RC4 Cipher"

_sensors, 2022, doi:10.3390/s22103844_

Round 1

Reviewer 1 Report

The author proposed a new binary HPSOEO method for RC4 cryptanalysis by combining particle swarm optimization (PSO) and equilibrium optimizer (EO). Experimental results show the proposed one is better than the others.

Strength:
- The result looks good.

Weakness:

- The proposed method is just a combination of two different approaches.

- The cryptography terms used in this paper are wired. For example, the "known plaintext attack (KPA)" is known as the "chosen plaintext attack" and the "ciphertext only attack" is known as the "chosen ciphertext attacks". If they mean the same thing, the authors should use formal names. Otherwise, formal definitions should be given.

- I think the authors are not familiar with the development of modern cryptography. Besides symmetric encryption and asymmetric encryption, there are also MAC, hash, ZKP, MPC, etc. You also use "public-key cryptography" in Introduction but "asymmetric cryptography" in Fig. 1.

- The advantages of "meta-heuristic optimization algorithms" are given without any description of the "meta-heuristic optimization algorithms".

- The title of Section 3 is confusing. "Methods" seem to be "our proposed techniques". However, it turns out to be the methods of existing approaches.

- In Section 3, different methods are given directly without any explanation. What these methods are used for? What are the notions of different symbols? The current version is unacceptable.

- The inputs/outputs of the algorithm are not given (Fig. 6).

- The HPSOEO algorithm is proposed directly without any explanation of why and how. It seems that you just combine two different approaches. Why use these two? Can one improve the disadvantage of the other one?

- It is well known that RC4 is not perfect. There is bias in the first 256 bytes of the output as well as the steam generator. Besides, RC4 is insecure when used with related keys. I'm not sure whether the authors consider these problems as there is no detailed description of the experiment.

- Figures are not clear. Many typos and grammatical mistakes make this paper hard to follow.

Reviewer 2 Report

The paper is surely interesting and from the optimisation-algorithms point-of-view is sound. However, the cryptanalytic part is, in my opinion, evaluated in the wrong way.

My main objection is that the authors seem to mistake the key of RC4 cipher, which is the value used in the key-scheduling algorithm and usually is 40 - 2048 bits in length, with the generated keystream of the RC4 cipher and its length differs according to the length of the plaintext to be encrypted. From the description of the experiments and the attack using the BHPSOEO, it is not clear to me, whether the particles of the population used in the optimization algorithm consisted of:

  1. candidates for the key (i.e. the value used in the keyschedule), or
  2. candidates for the keystream (i.e. the value generated by RC4 and XORed with the plaintext)

The second approach of course does not make any sense in the case of known-plaintext attack, because the attacker already knows the keystream. So I assume the authors meant the first approach, where their population consisted of key-candidates and they generated an RC4 keystream using each candidate and compared the generated keystream with the known one.

But in this case, their interpretation of the results (mainly written in the Conclusions) is wrong. Since they use a key of 32-bits (as stated e.g. in the Table 2) that means that any attacker who would try to find such a key, would have to brute force 2^32 values. But the authors compare their results with 2^64 and 2^128 respectively for ciphertexts of 64 and 128 bits. This is just wrong because no attacker would ever search for the keystream, in RC4 the attacker always searches for the key itself.

My belief that the authors do not differ the key and the keystream correctly is further strentghed by their fitness function (Eq. (21)), where they state that |K| is the length of the key and also state that the value of the fitness function is equal to one if the key has been guessed correctly. This is true in a very special case - if the length of the keystream and the key are equal, which is almost never the case in RC4. Usually, the keystream is longer than the key and then, the nominator of the Eq.(21) is longer than the denominator and therefore for a correct guess, the value of Eq.(21) would be larger than one. So it is my understanding that if the value of Eq.(21) should be 1 for the correct guess of the key, the denominator |K|  should stand for the length of the KEYSTREAM, not the length of the key.

To sum up, my main problem with the paper is a misleading distinction between the key and the keystream in the description of the attack, of the experiments and in the conclusion. This has to be rewritten to bring more clarity.

On the other hand, I find the results that using the hybrid approach should improve on the performance of PSO and EO very interesting, so the paper might be useful, but the cryptanalytic part simply has to be better described.

Also, the paper contains a fair share of typos, please find enclosed the pdf version of your paper with my comments on the typos. I have used the SumatraPDF programme for writing the comments.

Round 2

Reviewer 1 Report

Though the authors answer some of my questions, this paper still has many careless mistakes. I hope the authors carefully proofread the paper before submitting it.

1. The authors state, "the fundamental issue (of the block cipher) is that each piece of plain text is coded independently of the others" in the Introduction section, which is not true. This issue has been solved by using CBC, CTR, OCF, or CFB encryption mode. Almost no application uses EBC mode.

2. Inconsistent descriptions. V_j and X_j are used in the description of equations 1-3. But equations 1-3 use v_ij and x^t.

3. For equations 1-2, x^t means the product of t-many x. I think you mean x(t). (So does v^t)

4. Line 151. "Fig. 1 shows the flow chart for it." -> Fig. 3.

5. The inputs and outputs should be clearly stated in the algorithm.

6. I do not mean RC4 is not good and does not worth analysis. I mean, the known problems of RC4 seem not considered in the cryptanalysis of this paper. If a cryptanalysis approach leverages some known problems, it should easily outperform other approaches not knowing these problems. Accordingly, you should avoid such problems in the experiment. However, no description is given in the current version.

7. The figures are still not clear. You should use vectogram.

8. Typos.
"introduction" -> Introduction
"where the cryptanalyser 42 knows Ciphertext only" -> "ciphertext only"
"cipher text/plain text" and "ciphertext/plaintext" coexist

Author Response

Thank you very Thank you very much for reviewing my research paper for reviewing my research paper

Reviewer 2 Report

I think the authors have improved the paper and clarified the cryptanalytic part. I am satisfied with the current version of the paper and recommend it for publication.
